# OpenReview forum: "Private Attribute Inference from Images with Vision-Language Models"
_NeurIPS.cc/2024/Conference — NeurIPS 2024 poster_

### Official Review · Reviewer_T2ze · 2024-07-05

**Soundness:** 3
**Presentation:** 3
**Contribution:** 3
**Rating:** 6
**Confidence:** 4

**Summary:**

The paper presents a timely and important investigation into the privacy implications of Visual Language Models (VLMs) by evaluating their ability to predict sensitive personal information from images found online. The authors introduce a new dataset, which consists of:

1. Images sourced from posts on selected subreddits.
2. Eight categories of private attributes, including residence location, sex, age, and others.
3. 554 manually-labeled attributes, each assigned a "hardness score" reflecting the degree of reasoning and online search required for annotation.

The authors assess several VLMs (black-box), with GPT4-V achieving the highest average accuracy at 77.6%. The authors further explore the impact of prompt engineering and implement automated zooming through prompting.

**Strengths:**

1. The paper studies the crucial and under-explored issue of privacy risks associated with VLMs.
2. Unlike previous datasets that primarily rely on images of people, this dataset's annotated attributes only contain depictions of humans in 9.7% of cases. This allows for a more nuanced evaluation of VLMs' ability to infer sensitive information from subtler visual cues.
3. The paper is well-written, with clear documentation of the annotation and evaluation procedures.

**Weaknesses:**

1. The ground truth labels are created by annotators, which may introduce subjectivity and may not always reflect real-world truths.
2. The dataset is relatively small, and the absence of standard deviation reporting makes it difficult to assess the robustness of the results. If one labels another 500 attributes from another set of images collected using the same procedure, how would the accuracy change?
3. It would be better if the authors could provide alignment rates between different annotators.

**Questions:**

1. What is the time spent on filtering subreddits and on annotating the images separately? This could help reproducibility.
2. What are the estimated financial and time costs of human annotation compared to VLM-based annotation methods?

---

> ### Author Rebuttal · Authors · 2024-08-07
>
> We would like to thank the reviewer for their time and effort spent reviewing our paper, for their insightful and detailed feedback, and for their positive overall assessment of our work. We especially appreciate the reviewer’s acknowledgment of the criticality of the studied privacy issue and for their compliments of the constructed dataset and presentation of the work. Below, we address the reviewer’s questions and comments.
>
> **Q1: Could you please elaborate how the insights on the privacy issue discussed in the paper depend on the dataset?**
>
> Definitely. While we agree with the reviewer that the exact quantitative results presented in the paper may vary depending on the dataset, we believe that the key message of our paper is qualitative; namely that such privacy infringing inferences from ordinary images posted on anonymous forums are indeed possible at scale. We constructed our dataset in order to verify (or reject) this hypothesis, which we believe to have achieved, especially given the fact that our dataset is comprised of real-world images posted on an anonymous forum (Reddit), and as such, any inference success on our dataset has direct real-world implications. Unfortunately, due to our resource limitations and the expensive labeling procedure, we could not extend the dataset with more samples (we refer  to Q2 and Q3 below for more details). Nonetheless, we also believe that constructing large-scale and ideally public benchmarks for investigating and mitigating the privacy infringing inference capabilities for VLMs is interesting and important future work. However, to provide some robustness statistics of our results, we have (1) run our inferences three times at temperature 0.2 obtaining accuracies of 76.2%, 77.1%, 77.8% with GPT4-V, showing little variation, and (2) calculated a CI for our reported result in the paper at temperature 0.0 for GPT4-V: 77.6% +- 3.5% at confidence level 95%. We will include these additional results in the next revision.
>
> **Q2: “What is the time spent on filtering the subreddits and on annotating the images separately?”**
>
> Filtering the subreddits was based on fixed heuristics and as such it did not take any significant time. Annotating the images required around 1 week of full time work, i.e., roughly 40 hours.
>
> **Q3: “What are the estimated financial and time costs of human annotation compared to VLM-based annotation methods?”**
>
> As mentioned in Q2, the labeling of the dataset cost around 40 hours of human work. Using the hourly rates set by our institution (USD 35/h) this amounts to USD 1400 of labeling costs. At the same time, using the GPT4-V API, we spent less than USD 12 for inferences on the whole dataset, conducted in around 5 mins with further parallelization possible. As such, GPT4-V inferences are more than 100x cheaper and around 480x faster than human labeling, highlighting the concerning  scalability of VLM inferences vs. relying on human annotators for privacy infringing inferences. We thank the reviewer for raising this important point, and will include these numbers in the next revision of our paper.

---

> > ### Comment · Reviewer_T2ze · 2024-08-13
> >
> > Thank you for your response. I do not have other questions at this point.

---

### Official Review · Reviewer_g2WQ · 2024-07-05

**Soundness:** 2
**Presentation:** 2
**Contribution:** 2
**Rating:** 4
**Confidence:** 4

**Summary:**

The authors focus on the privacy risks of multimodal LLMs by demonstrating the personal information can be extracted from publicly accessible images, and leveraged by LLMs to infer sensitive details.

**Strengths:**

The intersection of privacy and multimodal AI is interesting, and the evaluation is strong, using both open and closed LLMs with vision capabilities. The paper is written well, and I appreciate the responsible disclosure.

**Weaknesses:**

I have two main issues with this paper:
- The authors claim that inference of personal details from public images constitutes a privacy violation. Many privacy researchers would disagree with this. It could be claimed that a person voluntarily posting an image online implies that any content of this image and any subsequent inference is an acceptable risk to them. If the image is posted involuntarily, then it's not the LLM inference that constitutes the privacy violation, but the unauthorised posting of the image. In other words, the fact that VLMs can extract personal information from images, while potentially aggravating to users, is not straightforwardly a privacy violation per se. There is no particular discussion about this, or the potential regulatory implications or mitigations for the issue.
- The fact that VLMs are capable of this type of inference is not particularly surprising to me, and the main takeaway is "a human with some time on their hands, access to a search engine and some open-source intelligence skills could do the same, but it'd take a more time". The only counter-argument here is scalability. There is no thorough discussion on potential incentives to perform this type of inference on an industrial scale.

In other words, I am hard-pressed to detect the strong, broad-interest contribution of this work beyond "VLMs can do XYZ efficiently", which is an addition to a long list of things that VLMs can do. Beyond this, the contributions are relatively low-impact: There is a some jailbreaking, and some engineering work to reveal details in the image by zooming automatically. For a NEURIPS paper, this is not a substantial enough contribution.

Beyond this, the evaluation on a single dataset (sourced from the internet, see below) which is also not very large, and lack of a comparison against human observers, narrows the impact of the work.

**Questions:**

- What are the takeaways of your works in terms of concrete recommendations for society, platform owners, regulators, and privacy researchers?
- Why was only a single dataset included? Can you certify that this dataset is representative of images encountered? I see that most of it is sourced from Reddit, are there any concerns of statistical bias from this method of selection?
- Why is there no evaluation against human observers? I think this would be the most relevant and interesting type of comparison here: a systematic evaluation against humans with OSINT expertise. Perhaps parts of a labelling workflow on a blinded group of evaluators could be leveraged for this task?
- What is the impact of hyperparameters? I see that the temperature was fixed. A more thorough evaluation could be useful here.
- What are the costs of executing these inferences in your work?
- It would be useful to have a bigger discussion around topics like e.g. uncertainty: The models can confidently hallucinate some personal property which misleads the "attacker" completely. How does this and similar phenomena affecting LLMs/VLMs interact with your findings?

**Limitations:**

Please see the "questions" on some of the scientific limitations.

I am a bit concerned about this study being executed without an institutional review board permission with the justification that "human subjects were not involved". Depending on jurisdiction, this can be contentious. You sourced images from Reddit, in my understanding, without asking for permission by the data owners, and conducted a scientific study on them. My understanding is that there was no statistical counselling about the representativeness of the dataset, or discussion with an ethics review board about the concerns arising from this study. In particular, the checklist states clearly that "The answer NA means that the paper does not involve crowdsourcing", and you answered "NA", but the dataset seems to be downloaded from Reddit, which (while not crowdsourcing in the strict sense) involves leveraging (in fact, attacking) human data without explicit consent, so I find the "NA" here problematic.

---

> ### Author Rebuttal · Authors · 2024-08-07
>
> We sincerely thank the reviewer for their time and effort spent reviewing our paper. We also appreciate the reviewer's acknowledgment of the studied problem being interesting, our evaluation and presentation. We hope our answers below address the reviewer's concerns.
>
> **Q1: Does the accurate, scalable, and automated inference of detailed personal attributes from images posted on pseudonymized platforms pose a privacy concern?**
>
> Yes, we strongly believe that the demonstrated inferences amount to a serious privacy threat. First of all, while it is indeed possible for humans to also infer such attributes, human inferences cost around 100x more money and 480x more time (we refer to Q2 and Q3 of Reviewer T2ze for detailed inference costs). As such, using VLMs for this task is a fundamental paradigm change. Notably, VLMs enable these inferences at a large and automated scale, questioning the privacy through obscurity assumptions one usually operates under on such platforms.
>
> Additionally, as detailed in the paper, key privacy regulations consider data that enables the inference of personal information protected data—something not commonly associated with snapshots of parking lots posted in largely non-privacy-concerning contexts.
>
> Further, our view that large-scale personal attribute inferences from online content **not intended** to be personally identifiable are shared by both the academic community and the public. This is highlighted by similar works [1] on text receiving widespread attention, including an ICLR spotlight and a privacy policy award. Further, similar issues are widely discussed in the community [2,3], as well as real-world applications [4].
>
> All in all, our work is the first to show that personal information is inferable at a large scale from inconspicuous images posted online, which is, in line with other reviewers, a concerning and relevant finding.
>
> **Q2: “What are the takeaways of your works in terms of concrete recommendations for society, platform owners, regulators, and privacy researchers?”**
>
> *Society*: We believe that our work can raise awareness among the public that sharing images, even on anonymous platforms where they thought to have taken enough care to obscure their identity, can be revealing of their personal information. We hope that, with awareness of this, users can adjust their online behavior and become more conscious of their privacy.
>
> *Platform Owners*: Anonymous platforms have to be aware of such risks, and may adjust privacy promises to their users accordingly. Educating their users of the full extent of the risks would be also desirable. At the same time, they could also take a conscious approach to make it harder to process their data at a large scale, obstructing scalable inferences.
>
> *Regulators*: We believe that in some jurisdictions, such as the EU, there are already strong regulatory protections for private data in place. As such, the important takeaway for regulators is to see how ubiquitous/heterogenous personal data really is.
>
> *Privacy Researchers*: We believe that for privacy researchers, in the face of these privacy risks posed by foundation models the key challenge to tackle is to develop both model side and user side mitigations.
>
> We thank the reviewer for this insightful question, and we will expand our paper in its next revision with a more detailed discussion.
>
> **Q3: Does the used evaluation dataset impact the qualitative conclusions drawn by the paper?**
>
> No, we believe that the key message of our paper is qualitative, namely that privacy-infringing inferences on real-world data at scale are possible, and this message is not impacted by the exact quantitative results obtained. Notably, the dataset that we have used for this work is sourced exactly where we see the examined threat: from real-world online forums. As such, any accurate inference on our dataset corresponds to a real-world privacy risk. This is also the reason why we have taken such care in constructing the dataset (labeling in-house) and not releasing it to the public.
>
> **Q4: Would an evaluation against humans with OSINT experience be possible?**
>
> While such a comparison would certainly be interesting, it is not the focus of the paper (see Q3). Further, as in our case, we already spent 100x more money and 480x more time on labeling than VLMs, a cost difference hard to overcome with domain experts. Finally, we believe that with our current real-world data, such an experiment would be ethically highly problematic, particularly when tasking outside individuals to infer personal data from real-world individuals.
>
> **Q5: What is the impact of the sampling temperature parameter on the quantitative inference results?**
>
> For this, we run GPT4-V on temperatures 0.0, 0.2, and 0.4, achieving 77.6%, 77.03% (average of three runs: 76.2%, 77.1%, 77.8%), and 77.3% accuracy, respectively. As such, reasonable temperature levels have little impact on the quantitative results. We will include these additional results in the next revision.
>
> **Q6: Could you please discuss how inference uncertainty reflects on your findings?**
>
> We agree with the reviewer that uncertainty estimates could help in real-world scenarios in case of inaccurate inferences. Qualitatively, similar to [5], we find that models are much less prone to hallucinations when they are tasked with directly reasoning about a given input (compared to, e.g., underspecified QA settings). Actively recognizing this provides an interesting avenue for future research in this area.
>
> **References**
>
> [1] R Staab et al. Beyond Memorization: Violating Privacy via Inference with Large Language Models. ICLR 2024.
>
> [2] Schneier, B. (2023, Dec). AI and Mass Spying. Schneier on security.
>
> [3] Eggsyntax. (2024, May). Language models model US. LessWrong.
>
> [4] Brewster, T. (2023, Nov) Chatgpt has been turned into a social media surveillance assistant.
>
> [5] R Staab et al : “Large Language Models are Advanced Anonymizers”, 2024; arXiv:2402.13846.

---

> > ### Comment · Reviewer_g2WQ · 2024-08-08
> > **Thank you and response to rebuttal.**
> >
> > Thank you for your rebuttal.
> >
> > > Q1: Does the accurate, scalable, and automated inference of detailed personal attributes from images posted on pseudonymized platforms pose a privacy concern?
> >
> > I recognise that there are varying valid viewpoints on this topic, and I will not hold this point against you.
> >
> > I have read the rest of your rebuttal and I did not find responses to all of the concerns I raised, in particular not the statistical validity and (especially) ethical considerations about the use of publicly sourced data and whether this study was guided by an institutional review board. The phrasing of your rebuttal actually amplifies some of my concerns ("such an experiment would be ethically highly problematic, particularly when tasking outside individuals to infer personal data from real-world individuals"). I am thus maintaining my current stance until this information is provided.

---

> > > ### Author Response · Authors · 2024-08-08
> > >
> > > First of all, we thank the reviewer for their response and for engaging in a discussion with us. We are also highly appreciative of their recognition of differing viewpoints on how and to what extent the presented inferences pose a privacy threat.
> > >
> > > Let us first answer an explicit question from the review that we have not answered directly in our rebuttal.
> > >
> > > **”What are the costs of executing these inferences in your work?”**
> > >
> > > Using the GPT4-V API, the cost of our inferences on the whole dataset lie below \\$12 and take around 5 minutes at our current level of parallelization (we believe that with a better implementation of parallelization, this can be further reduced). Note that since we have conducted this study, GPT4o has been made available, which has a significantly lower API cost. At the same time, creating the dataset (in-house human labeling) took us around 40 hours of work, which, using the hourly rates set by our institution (\\$35), amounted to a total labeling cost of \\$1400.
> > >
> > > Now, let us address the points explicitly raised in the reply.
> > >
> > > **”Why was only a single dataset included? Can you certify that this dataset is representative of images encountered? I see that most of it is sourced from Reddit, are there any concerns of statistical bias from this method of selection?”**
> > >
> > > We only used a single dataset for this study, as (1) there are no suitable available datasets in the community, as also elaborated in the paper; (2) constructing such datasets is very expensive, as discussed above; and (3) we were not able to find an equally suitable alternative data source to Reddit, which is also as directly representative of the actual examined inference threat.
> > >
> > > We collected the dataset by first heuristically filtering subreddits where we believed informative images are being posted (the subreddits are listed in Appendix D.4; visiting them on Reddit will give you a sense of the images included in the dataset). Note that these heuristics were mostly based on our intuition and did **not** involve any prior inference results. Further, the dataset only includes images where the labeler was able to infer at least one feature themselves. No other selection was made, especially, we **did not** adjust anything in the dataset after we have started with our inference experiments. As such, the dataset is not cherry-picked, and is representative of real-world images posted on anonymous forums from which personal attributes can be inferred. Note that the fact that we only include images in the dataset that can be labeled do not subtract from the representativeness of the threat. An adversary could simply first ask the VLM if it can infer anything, before proceeding to infer personal attributes; or simply ignore those answers of the model where no inferences have been made.
> > >
> > > Now, it is indeed true that images sourced from Reddit, especially from a select subset of subreddits is not representative of *all possible images* one may encounter online. However, it is also not the goal of the paper, as previously mentioned, to provide an exact and representative quantitative analysis across all online platforms of the image analysis capabilities of VLMs. Instead, the key message of the paper is qualitative; there exists a privacy threat enabled by the large-scale inferences of VLMs stemming from seemingly non-revealing images that people tend to upload on anonymous platforms under non-representative usernames. To deliver this message, we believe our dataset is extremely well suited, as it includes exactly these sorts of images posted on Reddit, a pseudo-anonymous forum. Each accurate inference we have made in a controlled setting, could also be made by an adversary with malicious intents on the exact same data points. This is also the reason why we have taken so much care when dealing with our dataset and publishing our results — we elaborate more on this in the next question.

---

> > > > ### Author Response · Authors · 2024-08-08
> > > >
> > > > **Please expand on the ethical considerations taken when conducting your study.**
> > > >
> > > > We value and share the reviewer’s concern when it comes to the ethical responsibilities connected with our work. This is why we have (1) used only images that were already included in at least one Reddit dump used for other research; (2) are publicly available; (3) kept the labeling in-house (*explicitly not sharing it with any outside party*); and (4) did not and will not release either the images used or the labels. Further, note that during our research, we did not deanonymize or identify any person whose data was in our dataset. Additionally, even though we kept all labeled data on premise, we still only included features into our study that were ethically representable, not trying to label and infer other, highly sensitive features such as race or mental health status.
> > > >
> > > > As such, we believe that the data we have used and the manner in which we have used is comparable to studies conducted on other research datasets sourced from natural persons, such as the famous Adult [1] dataset for tabular data research or several of the PAR and HAR datasets discussed in our paper. Hence, similarly as for research conducted on such datasets, we did not seek the guidance of an institutional review board for this study.
> > > >
> > > > Regarding our statement cited in the reviewer's response; we believe there is a stark difference in ethical implications between the way we have conducted our study and what it would mean to give access to our data to third-parties with OSINT experience for investigating the users behind the images.
> > > >
> > > > Finally, our choice of data source is in line with prominent research in this and related areas as well, see for instance [2-7].
> > > >
> > > > We hope to have been able to address the reviewer’s open points, and will include a discussion on the important points above in the next revision of our paper. In case there are any further questions or concerns, we are eager to further engage in this discussion.
> > > >
> > > > **References**
> > > >
> > > > [1] D Dua & C Graff. UCI machine learning repository. 2017.
> > > >
> > > > [2] R Staab et al. Beyond Memorization: Violating Privacy via Inference with Large Language Models. ICLR 2024.
> > > >
> > > > [3] Y Dou et al. Reducing Privacy Risks in Online Self-Disclosures with Language Models. ACL 2024.
> > > >
> > > > [4] J H Shen & F Rudzicz. Detecting anxiety on Reddit. CLPsych 2017.
> > > >
> > > > [5] F Rangel et al. Overview of the Author Profiling Task at PAN 2013. 2013.
> > > >
> > > > [6] F Rangel et al. Overview of the 5th Author Profiling Task at PAN 2017: Gender and Language Variety Identification in Twitter. 2017
> > > >
> > > > [7] F Rangel et al. Overview of the 6th Author Profiling Task at PAN 2018: Multimodal Gender Identification in Twitter. 2018.

---

> > > > > ### Comment · Reviewer_g2WQ · 2024-08-09
> > > > >
> > > > > Thank you for clarifying these points.

---

> > > > > > ### Author Response · Authors · 2024-08-09
> > > > > >
> > > > > > We thank the reviewer for the insightful discussion and for raising their score! If the reviewer still deems that some of their points have been insufficiently addressed, or has any other open points, please let us know, we are eager to continue this fruiful discussion.

---

### Official Review · Reviewer_dw5b · 2024-07-12

**Soundness:** 3
**Presentation:** 3
**Contribution:** 3
**Rating:** 7
**Confidence:** 4

**Summary:**

The paper performs a novel analysis privacy-leakage due to modern multimodal VLMs. Specifically, they show that modern VLMs, when correctly promoted, can infer sensitive information from seemingly innocuous images. They curate a dataset of images containing clues to sensitive information and query a few SOTA models using these [image, prompt] pairs, and show that the models are capable of inferring eight sensitive attributes from the images. The inference ability strengthens with the quality of models, which raises concerns about online privacy.

**Strengths:**

- Very interesting, timely and practically relevant privacy analysis of VLMs
- The work raises appropriate concerns about privacy of online images
- Paper is very well-written and easy to follow

**Weaknesses:**

- The proposed attack although important seems quite easy to defend against
- The paper should evaluate at least a couple of  simplest of defenses, e.g., SFT.

**Questions:**

I really like the work due to its relevance to practice and shoutout to the authors for that! I have a few questions:
- The proposed attack seems super simple: it’s basically jailbreaking VLMs to elicit harmful responses. This is known type of an attack that is extensively studied, e..g, for LLMs. How will this attack work if a proprietary model provider does SFT using aligned data?

**Limitations:**

See questions above.

---

> ### Author Rebuttal · Authors · 2024-08-07
>
> First, we would like to sincerely thank the reviewer for their time and effort spent reviewing, their insightful comments, and for their highly favorable assessment of our work. We are especially thankful for the reviewer’s acknowledgement of the relevance and timeliness of the presented privacy threat. Below, we address the reviewer’s comments and questions.
>
> **Q1: Would it be easy to defend against the presented inference-based privacy attack?**
>
> Crucially, independently of how easy it would be to defend against such privacy inferences on the model provider’s side, the fact that these privacy violations are possible with current available models is concerning. Especially, since open-source models are also capable of highly accurate inferences, meaning that even if model providers would make preventative adjustments, these previously uploaded open-source models would still be available to conduct privacy-infringing inferences.
>
> Further, while it is hard to exactly anticipate the difficulty of provider side defenses, we believe that it would be a difficult undertaking to try to prevent privacy infringing inferences while maintaining utility on other tasks. We believe so, as we have observed (as also stated in the paper) that the capability of the models to conduct such inferences is highly correlated with their general capabilities. Additionally, we know that some of the examined models (e.g., GPT4-V) already are supposed to be aligned against such inferences, yet we were able to harness these models without too much effort put into jailbreaking them. In fact, for this rebuttal, we tested our attack on the newer GPT4-o model, which tends to score higher in alignment benchmarks than GPT4-V [1], and achieved even higher accuracy (80.7%); showing that alignment against such inferences is at the moment not strong enough.
>
> Nonetheless, as also argued in the paper, we strongly believe that developing adequate defenses, both on the providers’ side and on the users’ side is crucial going forward in order to enable people to exercise their right to privacy even in the age of VLMs. In this regard, we make the first important step into this direction by pointing out, and systematically evaluating the inference-based privacy risks these VLMs pose; hopefully raising awareness in the broader privacy and machine learning community.
>
> **Q2: Could SFT be an effective method to defend against the attack?**
>
> We believe that both SFT, and especially preference tuning, such as PPO or DPO could be promising strategies on the model provider’s side to decrease the likelihood of a model answering a privacy infringing inference query. However, conducting such experiments is beyond our means unfortunately as (1) even the examined well-performing open-source models are extremely large to handle for training, and (2) this would require vast amounts of training data that is even more expensive to obtain (please see our responses to Q2 and Q3 of Reviewer T2ze for more details). Nonetheless, we agree with the reviewer and believe that it is an important avenue both for future scientific work and for commercial providers to explore the possibilities of preventing privacy-infringing inferences through finetuning or preference tuning. As part of our responsible disclosure we therefore directly contacted all LLM providers used during this study.
>
> **References**
>
> [1] Z Ying et al. Unveiling the Safety of GPT-4o: An Empirical Study using Jailbreak Attacks. arXiv 2024.

---

### Official Review · Reviewer_zT6i · 2024-07-12

**Soundness:** 2
**Presentation:** 3
**Contribution:** 3
**Rating:** 5
**Confidence:** 4

**Summary:**

The paper proposes a new privacy attack with VLMs where the model is queried with an image and the goal is to predict the private attributes such as place and gender of the person not shown in the image.

**Strengths:**

- Good experimental evaluations
- Clear problem formulation
- Practical automated attack using VLMs

**Weaknesses:**

Overall, I like the evaluation of the attack, although I feel it is missing a baseline comparison. For instance, how likely is it to predict the sensitive attributes like age, gender, etc. (without access to the VLM) using pure correlations in the data. If there is a pipeline for automatically annotating the important objects in the image, then a Bayesian model or a neural network can be trained to capture the correlations to the target attribute. This is also indicative from the experiments where the presence of partial human identifiers boosts the attack success. This type of correlation can be captured by a baseline model.

**Questions:**

Can you compare the effectivness of VLMs w.r.t. the baseline model that captures the general correlations between objects in the images?

**Limitations:**

The authors show a practical attack using modern VLMs where the adversary can infer sensitive attributes about an image by querying these models. While authors do not release their VIP dataset citing ethical concerns, I feel they need to acknowledge that an adversary can replicate their attack in practice, and should discuss more on the defenses against such attacks.

---

> ### Author Rebuttal · Authors · 2024-08-07
>
> We thank the reviewer for their time and efforts spent reviewing and for their insightful feedback. We are especially appreciative of the reviewer’s recognition of our attack’s novelty and our extensive experimental evaluation. Below, we address the reviewer’s comments and questions.
>
> **Q1: Would it be possible to construct a baseline method relying on classical supervised learning techniques?**
>
> Fundamentally, any baseline method relying on supervised learning would require a specialized training dataset on which it has to be trained. As such, mounting a privacy attack using such methods would require significantly stronger assumptions on the adversary. At the same time, and as we argue in our paper, VLMs pose a high privacy risk through inference exactly because of the low barrier of entry for adversaries.
>
> While we agree with the reviewer that such a baseline evaluated directly on our dataset would provide an interesting anchor point for our results, unfortunately, due to the highly diverse and challenging nature of the domain this is not feasible.
>
> To underline this argument, take the following examples of how the VLM inferences have been made: (1) an office room, where among many of the items there is one with a University of Florida logo; (2) the interior of a New York public library; (3) sunset, with a view of a few buildings in Denver, in the foreground an open bottle of German beer; (4) just a picture of an at home office room, stylistic elements indicating the (likely) sex of the author.
>
> Each of the above inferences relies on specific task-dependent knowledge that would need thousands of data points to learn in a supervised setting. As labeling even the few hundred examples in our dataset is considerably expensive (~USD 1400, please see our responses to Q2 and Q3 of Reviewer T2ze for more details), obtaining such a dataset for each possible inference task is not feasible. Clearly, in order to excel at such inferences, the VLMs have to rely on the vast world knowledge obtained during unsupervised pre-training both on image and text data.
>
> Further, the closest direct comparison on a related domain that we can obtain is comparing the performance of VLMs against “traditional” machine and deep learning methods on HAI and PAI datasets, where, as mentioned also in the paper, visual foundation models have shown promising performance over prior methods [1-3]. As such, we also believe that even if one could construct supervised methods for some of the inference types, VLMs would still prevail.
>
> **Q2: Could you please expand the discussion on the threat of adversaries replicating your attack and on possible defenses?**
>
> Certainly. In general, we share the reviewer’s concern, and it was our main motivation for conducting this study to raise awareness about the inference-based privacy threat VLMs pose such that the community can (1) act against it by mitigating the issue, and (2) people can adjust their online practices in the face of this privacy threat. In line also with our concern, before making a copy of this study available anywhere, we first informed the key entities that produced the examined VLMs about our findings as part of a responsible disclosure. All in all, in line with the common stance in the privacy community, we believe that privacy through obscurity is not robust; which is why we think it is important to raise and systematically evaluate this issue in the form of this study.
>
> Regarding potential mitigations against the attack, as already discussed in Section 6 of the paper, we believe that both internet users and VLM providers can act towards reducing the privacy risks posed by VLMs. From the providers’ side, our findings could serve to strengthen the safety alignment of the models, specifically targeting privacy infringing inferences. From the users’ side, we hope for anonymization tools that could extend to images, by, e.g., removing revealing clues from images using generative modeling.
>
> Nonetheless, most importantly, we believe that the first, and most crucial step, towards responsible and privacy preserving practices of VLM development and use is awareness of the full extent of possible privacy issues these models may pose. Here, our paper focuses on raising and demonstrating the issue of VLMs enabling privacy violating inferences at a large scale.
>
> **References**
>
> [1] X Cheng et al. A simple visual-textual baseline for pedestrian attribute recognition. IEEE TCSVT 2022.
>
> [2] M Castrillón-Santana et al. Evaluation of a visual question answering architecture for pedestrian attribute recognition. CAIP 2023.
>
> [3] X Wang et al. Pedestrian attribute recognition via clip based prompt vision-language fusion. arXiv 2023.

---

> > ### Comment · Reviewer_zT6i · 2024-08-10
> >
> > Thank you for your clarifications. After reading the rebuttal and all the other reviews, I have decided to raise my score.

---

> ### Author Response · Authors · 2024-08-11
>
> We thank the reviewer for raising their score and if they have any remaining questions or concerns, we are eager to engage in further discussion until the end of the author-reviewer discussion period.

---

### Author Rebuttal · Authors · 2024-08-07

First of all, we would like to thank all reviewers for their time and efforts spent on reviewing our paper, and for their insightful, constructive, and valuable comments. We are especially appreciative of several reviewers’ acknowledgement of the relevance, practicality, and cruciality of the examined threat; as well as of the recognition of our extensive experimental evaluation.

We address the reviewers’ questions and comments in individual responses below, and are looking forward to a fruitful discussion.

---

### Comment · Area_Chair_KSNb · 2024-08-07

Hi all,

The author rebuttal period is now officially over. Could you please read over the rebuttal carefully and discuss with the authors if you have remaining questions? If not, please acknowledge that you have read the rebuttal and have come to a conclusion. Thank you!

---

### Decision · Program_Chairs · 2024-09-25

**Decision:**

Accept (poster)

**Comment:**

This paper studies the problem of private attribute inference using frontier vision-language models. Although there is prior work on similar studies for text-only LLMs, this is the first study that includes the vision modality. The authors construct a dataset systematically and evaluates frontier VLMs such as GPT-4V, Gemini Pro and LLaVA-NeXT, and finds that these models can accurately infer private attributes such as sex, age, location, income, etc. even when the image does not contain these attributes explicitly.

Reviewers generally agree this is a timely and important study and find the result meaningful. There are some minor concerns such as missing baseline using weaker models or a human evaluator, and missing evaluation of simple defenses such as alignment. After author rebuttal and discussion period, reviewers generally agree this paper's merits outweigh its weaknesses, and therefore AC recommends acceptance.